# A Pair of Co^II^ Supramolecular Isomers Based on Flexible Bis-Pyridyl-Bis-Amide and Angular Dicarboxylate Ligands

**DOI:** 10.3390/molecules25010201

**Published:** 2020-01-03

**Authors:** Pradhumna Mahat Chhetri, Ming-Hao Wu, Chou-Ting Hsieh, Xiang-Kai Yang, Chen-Ming Wu, En-Che Yang, Chih-Chieh Wang, Jhy-Der Chen

**Affiliations:** 1Department of Chemistry, Amrit Campus, Tribhuvan University, Kathmandu 44600, Nepal; mahatp@gmail.com; 2Department of Chemistry, Chung Yuan Christian University, Chung-Li 320, Taiwan; chriswu321@hotmail.com (M.-H.W.); jack63096309@hotmail.com (C.-T.H.); xiangkaishulin@gmail.com (X.-K.Y.); 3Department of Chemistry, Fu Jen Catholic University, New Taipei City 24205, Taiwan; fightpf@gmail.com; 4Department of Chemistry, Soochow University, Taipei 111, Taiwan

**Keywords:** coordination polymer, supramolecular isomer, coordination chemistry, entanglement

## Abstract

Thermal reactions of cobalt(II) salts with flexible *N,N′*-bis(pyrid-3-ylmethyl)adipoamide (**L**) and angular 4,4′-sulfonyldibenzoic acid (H_2_SDA) in H_2_O and CH_3_OH afforded a pair of supramolecular isomers: [Co_2_(**L**)(SDA)_2_], **1**, and [Co_2_(**L**)(SDA)_2_]⋅CH_3_OH⋅H_2_O, **2**. The structure of complex **1** can be simplified as a one-dimensional (1D) looped chain with **L** ligands penetrating into the middles of squares, forming a new type of self-catenated net with the (4^2^⋅5^4^)(4)_2_(5)_2_ topology, whereas complex **2** displays a 2-fold interpenetrated 2D net with the rare (4^2^⋅6^8^⋅8⋅10^4^)(4)_2_-2,6L1 topology. While both complexes **1** and **2** display antiferromagnetism with strong spin-orbital coupling, the antiferromagnetism of **2** is accompanied by a cross-over behavior and probably a spin canting phenomenon.

## 1. Introduction

The preparation and structural characterization of coordination polymers (CPs) with manipulable networks by judicial design continue to be an important point of contest in the research fields of crystal engineering, presumably due to their fascinating structural types and various potential applications in gas storage, catalysis, ion exchange, photoluminescence and magnetism [1,2,3,4,5]. The structural diversity of CPs is subject to various factors involving the identity of the metal ion, ligand and counterion, metal-to-ligand ratio, temperature, and solvent system. The supramolecular isomers of CPs are those that show identical chemical compositions but differ in the structural types of the core structures. Their structural types can also be altered by using different experimental conditions [6,7,8,9,10]. Although a few supramolecular isomers based on both dicarboxylate and bis-pyridyl ligands have been reported [8], it remains a challenge to elucidate the structure–ligand relationship and thereby the intrinsic properties.

Many CPs reported so far are quite attractive due to the formation of independent motifs entangled in different ways, such as interpenetration, polycatenation and self-catenation, all of which can be ascribed to the presence of large free voids in a single network. Entanglements may thus happen to enhance the packing efficiency of a crystal structure [11,12]. The structures of CPs based on flexible bis-pyridyl-bis-amide (bpba) ligands are less foreseeable, most probably due to the occurrence of supramolecular isomerism [13]. The flexible bpba may show various ligand conformations in the CPs and are considered crucial to the formation of entangled structures [13].

To explore the influence of angular dicarboxylate ligands on the entanglement of flexible bpba-based Co^II^ CPs and to elucidate their structure–ligand relationship, as well as to inspect the geometric effect on their magnetic properties, we have investigated the reactions of Co^II^ salts with *N,N′*-bis(pyrid-3-ylmethyl)adipoamide (**L**) and 4,4′-sulfonyldibenzoic acid (H_2_SDA) (Scheme 1). Herein, we report the syntheses, structures and magnetic properties of a pair of Co^II^ supramolecular isomers, [Co_2_(**L**)(SDA)_2_], **1**, and [Co_2_(**L**)(SDA)_2_]⋅CH_3_OH⋅H_2_O, **2**.

## 2. Results and Discussion

### 2.1. Synthesis

The combination of a flexible bpba ligand and an angular dicarboxylate is suitable for the formation of entangled CPs [13]. The *N,N′*-bis(pyrid-3-ylmethyl)adipoamide (**L**) ligands were thus reacted with the Co^II^ salts and 4,4′-sulfonyldibenzoic acid (H_2_SDA) to prepare CPs **1** and **2**, showing a new type of self-catenated net with the (4^2^⋅5^4^)(4)_2_(5)_2_ topology, and a 2-fold interpenetrated 2D net with the rare (4^2^⋅6^8^⋅8⋅10^4^)(4)_2_-2,6L1 topology, respectively, vide infra. The absorptions in the IR spectrum of **1** at 3444 cm^−1^ can be assigned to N–H stretching, and the one at 1557 cm^−1^ is due to amide C=O stretching, while the peaks at 1294 and 1160 cm^−1^ can be ascribed to S=O stretching, indicating the existence of **L** and SDA^2−^ ligands, respectively. Similarly, the characteristic peaks of **2** at 3401 cm^−1^ are due to N–H stretching, and the one at 1538 cm^−1^ can be assigned to amide C=O stretching, while the two peaks at 1296 and 1159 cm^−1^ can be ascribed to S=O stretching.

The different structural types of **1** and **2** were most probably directed by the solvents H_2_O and CH_3_OH, as well as the anions of the Co^II^ salts OAc^−^ and BF_4_^−^, which may exhibit a template effect on the structures of **1** and **2**, respectively.

### 2.2. Structure of ***1***

The structure of complex **1** was solved in the space group *P*ī. Figure 1a depicts a structure showing the coordination environment about the dinuclear Co^II^ metal centers. The Co–Co distance of 2.929(6) Å is too long to form a bond [14]. Both Co and Co(A) atoms are six-coordinated by one pyridyl nitrogen atom of one **L** ligand [Co–N = 2.058(2) Å], and five oxygen atoms of four *μ*_4_-*κ*^2^,*κ*^1^,*κ*^1^,*κ*^1^-SDA^2−^ ligands [Co–O = 2.020(2) Å–2.263(2) Å], resulting in distorted octahedral geometries. Figure 1b shows that the dinuclear Co_2_^4+^ ions are bridged by the SDA^2−^ ligands to form a 1D looped chain, with the flexible **L** ligands penetrating into the middles of the loops. The metal–metal separation between two dinuclear units is 13.28 Å. If the Co atoms are considered as 5-connected nodes, the SDA^2−^ ligands as 4-connected nodes and the **L** ligands as linkers, the structure of **1** can be regarded as a 1D net with the (3⋅4^5^)(3^2^⋅4^6^⋅5^2^)-4,5C11 topology (Figure 1c), determined using ToposPro [15]. If the shapes of the **L** ligands are taken into account and considered as two 2-connected nodes, the structure of **1** can be simplified as a (2-c)(4-c)(5-c), 3-nodal net with the (4^6^⋅5^2^⋅7^2^)(4^6^)(5) topology (Figure 1d). Moreover, if the dinuclear Co_2_(μ-COO)_2_ units are considered as 6-connected nodes, the structure of **1** can be further simplified as a (2-c)_2_(2-c)_2_(6-c), 3-nodal net with the (4^2^⋅5^4^)(4)_2_(5)_2_ topology (Figure 1e).

The structure of complex **1** is worthy of further mention. It has been clearly delineated that “self-catenated nets are single nets that exhibit the peculiar feature of containing shortest rings through which pass other components of the same network” [16]. The catenation of the self-catenated nets can thus be identified by the presence of edges that thread a ring [12]. Since complex **1** forms a 1D looped chain with one flexible **L** ligand penetrating into the middle of each of the loops, it can be regarded as a new type of 1D self-catenated net with a simple structure. This net is in marked contrast to that of {[Cu(L^1^)(1,4-pda)]·2H_2_O}_n_ (L^1^ = *N,N’*-di(3-pyridyl)suberoamide; 1,4-H_2_pda = 1,4-phenylenediacetic acid), which is a 1D entangled self-catenated net [12].

### 2.3. Structure of ***2***

The structure of complex **2** was solved in the space group *P*ī. Figure 2a depicts a structure showing the coordination environment about the dinuclear Co^II^ centers with a Co–Co bond distance of 2.831(7) Å, indicating no metal–metal bond formation [14]. Both Co(1) and Co(2) are 5-coordinated by one pyridyl nitrogen atom of one **L** ligand [Co–N = 2.052(3) Å], and four oxygen atoms of four *μ*_4_-*κ*^1^,*κ*^1^,*κ*^1^,*κ*^1^-SDA^2−^ ligands [Co–O = 2.015(3) Å–2.044(3) Å], resulting in a distorted square pyramidal geometry. Figure 2b shows that the Co^II^ ions are bridged by four carboxylate groups of the SDA^2−^ ligands to form 1D infinite looped chains with dinuclear paddlewheel units, which are further linked by the **L** ligands to form a 2D layer. If the dinuclear units are considered as 6-connected nodes, the SDA^2−^ ligands as 2-connected nodes and the **L** ligands as linkers, the structure of **2** can be simplified as a rare 2-fold interpenetrated 2D net with the (4^2^⋅6^8^⋅8⋅10^4^)(4)_2_-2,6L1 topology, as shown in Figure 2c. The water and methanol solvent molecules are located in the cavities of the interpenetrated framework. The water molecules interlink the independent nets through O–H---O hydrogen bonds to the amide oxygen atoms (O–H = 1.99 Å; ∠O–H---O = 153.7°) of one net and to the carboxylate oxygen atoms of the other net (O–H = 2.44 Å; ∠O–H---O = 145.5°), while the methanol molecules link two water molecules (Appendix A).

The first complex of the same structural type was found in the complex [Co(L^2^)_0.5_(MBA)]_n_ [L^2^ = *N,N’*-di(4-pyridyl)suberoamide; H_2_MBA = diphenylmethane-4,4′–dicarboxylic acid] [17]. The structure of **2** is in marked contrast to those observed in {[Co_2_(L^2^)(OBA)_2_] ⋅ 7CH_3_OH}_n_ [H_2_OBA = 4,4′–oxybis(benzoic acid)] [17] and {[Ni_2_(L^2^)(SDA)_2_]⋅6H_2_O}_n_, [18] which show 2D layers that catenate to each other to form 2D → 3D inclined polycatenation networks.

### 2.4. Ligand Conformations

The ligand conformations of the bpba ligands can be determined by following the reported procedures [13]: (a) If the C-C-C-C torsion angle (θ) of the backbone carbon atoms is 180 ≥ θ > 90° and 0 ≤ θ ≤ 90°, the ligand shows the A and G conformations, respectively. (b) The *cis* or *trans* arrangement is determined according to the relative orientation of the C=O (or N–H) groups. (c) The *anti*-*anti*, *syn*-*anti* and *syn*-*syn* arrangements are verified on the basis of the different orientations of the pyridyl nitrogen atoms. In accordance with this descriptor, all of the **L** ligands of complexes **1** and **2** adopt the AAA-trans *anti*-*anti* conformation (θ = 177.9, 180 and 177.9° for **1**; and 179.1, 180 and 179.1° for **2**), indicating that, instead of the ligand conformation, the structural diversity of **1** and **2** is subject to the co-crystallization of the solvent molecules.

### 2.5. Thermal Properties

Thermal gravimetric analyses (TGA) were carried out to examine the thermal decomposition of **1**–**2**. The samples were heated up to 900 °C under 1 atm pressure in a nitrogen atmosphere with a heating rate of 10 °C min^−1^ (Figure 3). The TGA curve of **1** shows that no weight loss can be observed up to 300 °C, and the weight loss of 83.6% (calculated: 88.8%) at 300–836 °C can be ascribed to the decomposition of SDA^2−^ and **L** ligands. The TGA curve of **2** indicates that the first weight loss of 4.2% occurred up to 120 °C, which is presumably due to the removal of co-crystallized H_2_O and CH_3_OH (calculated: 4.5%), while the weight loss of 81.0% (calculated: 84.7%) at 350–900 °C can be ascribed to the decomposition of SDA^2−^ and **L** ligands. Based on the starting decomposition temperatures of the ligands of **1** (300 °C) and **2** (350 °C), it can be proposed that the 2-fold interpenetrated 2D network is relatively stable compared to the 1D self-catenated network. Moreover, the powder X-ray diffraction (PXRD) pattern of **2** heated at 250 °C for two hours is quite similar to that of the original **2**, indicating that no structural transformation occurred upon solvent removal (Figure 4).

### 2.6. Magnetic Properties of ***1*** and ***2***

The test samples were obtained by carefully grinding single crystals of **1** and **2**, and the consistency of the crystal structures of the ground samples was checked by using PXRD. Appendix A show that the calculated pattern is quite consistent with the observed one, indicating that the structures of **1** and **2** remain stable during grinding. The ordering temperatures of the samples were then determined by conducting χ-T experiments on the SQUID system with a 1 kOe external magnetic field.

The dc magnetic susceptibility measurements of complexes **1** and **2** were conducted on polycrystalline samples by applying a 1000 G magnetic field in the temperature range 2–300 K. The χ_M_T vs. T and 1/χ vs. T plots of **1** are shown in Figure 5a,b, respectively. The χ_M_T value of 5.55 cm^3^ K/mol at 300 K is significantly larger than the spin-only value of 3.74 cm^3^ K/mol for two isolated cobalt(II) centers, each with three unpaired electrons and g = 2.0. This can be mainly due to the spin-orbit coupling of the high-spin Co^2+^ ions. The χ_M_T values decrease gradually to 4.67 cm^3^ K/mol at 100 K, then drop abruptly to 0.12 cm^3^ K/mol at 2 K, indicating the existence of an antiferromagnetic coupling as well as a strong spin-orbital coupling. It is thus difficult to give a precise analytical expression for the magnetic properties of **1** by using normal procedures. However, Rueff and his coworkers [19] have successfully given a phenomenological approach for the Co^II^ system, χ_M_T = A exp(−E_1_/kT) + B exp(−E_2_/kT), where A + B gives the Curie constant, E_1_ is the spin-orbital coupling constant and E_2_ is the antiferromagnetic coupling interactions. By using this equation, the results involving (A + B) = 6.13 cm^3^ K/mol, E_1_/k =110.4 K and E_2_/k = −J/2 = 16.1 K can be obtained, which are confirmed by following that the standard Curie constant is 6.25 cm^3^ K/mol, while an octahedral Co^II^ ion usually gives a spin-orbital coupling around 100.0 K. We therefore conclude that this set of parameters give a reasonable description for **1**.

The χ_M_T vs. T plot of complex **2** is shown in Figure 6a. The χ_M_T value is 6.33 cm^3^ K/mol at 300 K, which is significantly larger than that of complex **1**. It is also larger than the spin-only value of 3.74 cm^3^ K/mol for the two uncoupled Co^2+^ centers with S = 3/2 and g = 2, indicating strong spin-orbital coupling interactions. The χ_M_T values drop significantly to 0.89 cm^3^ K/mol at 25 K and then show a spike feature at 5 K, which is normal for antiferromagnetically coupled Co^II^ ions and probably indicates a spin canting phenomenon [20]. However, our attempt to analyze the magnetic properties of **2** by following Rueff’s phenomenological approach afforded no acceptable results. The (A + B) parameter as high as 13.3 cm^3^ K/mol is roughly double that of the standard Curie constant of 6.25 cm^3^ K/mol. A plot of 1/χ_M_ vs. T (Figure 6b) shows the significant deviation from Curie–Weiss law, and a comparison with those of typical Co^II^ spin cross-over complexes such as [Co(terpy)_2_]X_2_ [21] and [Co_3_(μ_3_-dpa)_4_Cl_2_] [22] implies the spin cross-over phenomenon in **2**. Because the magnetic behavior of **2** is complicated by strong antiferromagnetic coupling, a reasonable quantitative analysis for the high-low spin barrier is not reachable. However, on the basis of Figure 6a,b, we suggest that complex **2** displays a strong antiferromagnetism, accompanied with a spin cross-over behavior and probably a spin canting phenomenon.

## 3. Materials and Methods

### 3.1. General Procedures

Elemental analyses were obtained from a HERAEUS VaruoEL analyser (Elementar Americas Inc., Ronkonkoma, NJ, USA). The IR spectra (KBr disk) were recorded on a Jasco FT/IR-460 plus spectrometer (JASCO, 28600 Mary’s Court City, MD, USA). Thermal gravimetric analyses measurements were carried out on a TG/DTA 6200 analyzer (Seiko Instruments Inc., Tokyo, Japan). PXRD measurements were performed using a PANalytical PW3040/60 X’pert Pro diffractometer (PANalytical, EA Almelo, The Netherlands).

### 3.2. Materials

The reagents Co(CH_3_COO)_2_·4H_2_O, Co(BF_4_)_2_·6H_2_O and 4,4′-sulfonyldibenzoic acid (H_2_SDA) were purchased from Aldrich Chemical Co (St. Louis, MO, USA). The ligand *N,N’*-bis(pyrid-3-ylmethyl)adipoamide (**L**) was prepared by using modified procedures for bpba ligands [13,23].

### 3.3. Preparations

#### 3.3.1. Synthesis of [Co_2_(**L**)(SDA)_2_], **1**

Co(CH_3_COO)_2_·4H_2_O (0.05 g, 0.20 mmol), H_2_SDA (0.06 g, 0.20 mmol) and **L** (0.032 g, 0.10 mmol) in 10 mL H_2_O were placed in a 23 mL Teflon-lined stainless container, which was sealed and heated at 120 °C for 2 days and then cooled down slowly to room temperature. Red crystals were formed and then collected, washed with diethyl ether and dried under a vacuum. Yield: 0.067 g (64%, based on Co). Anal. calcd. for C_46_H_38_S_2_N_4_O_14_Co_2_ (MW = 1052.78): C, 52.47; N, 5.32; H, 3.63%. Found: C, 51.97; N, 4.86; H, 4.03%. Selected IR(cm^−1^): 3444(m, N–H), 3289(m), 1625(s, C=O), 1557(s, amide C=O), 1402(s), 1294(m, S=O),1160(s, S=O), 1100(m), 780(m), 740(m), 622(m).

#### 3.3.2. Synthesis of [Co_2_(**L**)(SDA)_2_]⋅CH_3_OH⋅H_2_O, **2**

This complex was prepared by following similar procedures to those carried out on **1**, except that Co(BF_4_)_2_·6H_2_O (0.086 g, 0.20 mmol) and 10 mL CH_3_OH were used. Blue-green crystals were derived. Yield: 0.034 g (30%, based on Co). Anal. calcd. for C_47_H_44_S_2_N_4_O_16_Co_2_ (MW = 1102.86): C, 51.18; N, 5.08; H, 4.02% Found: C, 51.13; N, 5.17; H, 4.48%. Selected IR(cm^−1^): 3400(s, N–H), 1635(s, C=O), 1538(m, amide C=O), 1400(s), 1296(s, S=O), 1159(s, S=O), 1103(m), 741(m), 622(m), 506(m).

### 3.4. X-Ray Crystallography

Single crystals of **1** and **2** that were suitable for structural determination were obtained from hydrothermal and solvothermal reactions, respectively, and their diffraction data were collected on a Bruker AXS SMART APEX II diffractometer, using graphite-monochromated MoK_α_ radiation with λ_α_ = 0.71073 Å. The reflections of **1** and **2** that were collected were reduced and corrected by using Lorentz-polarization [24], and their structures were solved by using the direct method in the SHELXTL-97 program [25], both of which are well-established computational procedures. The solutions first afforded the positions of some of the heavier atoms, including the cobalt atom. The remaining atoms were found in a series of difference Fourier maps and least-square refinements, while the hydrogen atoms were added by using the HADD command. The methanol carbon atom is disordered such that two orientations can be found. Table 1 lists the crystal parameters and structural refinement results.

## 4. Conclusions

A pair of Co^II^ CPs, [Co_2_(**L**)(SDA)_2_]_n_, **1**, and [Co_2_(**L**)(SDA)_2_⋅CH_3_OH⋅H_2_O]_n_, **2**, constructed from flexible *N,N′*-bis(pyrid-3-ylmethyl)adipoamide (**L**) and angular 4,4′-sulfonyldibenzoic acid (H_2_SDA) have been synthesized under hydrothermal and solvothermal conditions, respectively. The structural types of **1** and **2** are directed by the solvents and/or the anions of the reaction systems. The **L** ligands of **1** penetrate into the middle of squares formed by dinuclear Co_2_^4+^ ions bridged by SDA^2^^−^ ligands, resulting in a new 1D self-catenated net with a simple structure, and the 2D layers of **2** entangle each other to form a rare 2-fold interpenetrated 2D net with the (4^2^,6^8^,8,10^4^)(4)_2_-2,6L1 topology. We have verified that the combination of flexible bpba and angular dicarboxylate ligands may afford interesting entangled CPs. Complexes **1** and **2** display antiferromagnetism with strong spin-orbital coupling, while the antiferromagnetism of **2** is accompanied by a cross-over behavior and probably a spin canting phenomenon. This study provides an insight into understanding the magnetostructural relationship in a pair of Co^II^ supramolecular isomers based on bpba and angular dicarboxylate ligands.

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
