# Peer review of "A Pair of CoII Supramolecular Isomers Based on Flexible Bis-Pyridyl-Bis-Amide and Angular Dicarboxylate Ligands"

_molecules, 2020, doi:10.3390/molecules25010201_

Round 1

Reviewer 1 Report

In this paper the authors describe the synthesis, crystal structure and magnetic properties of two new Co-based compounds.

Concerning the crystal structure description, the figure must have the axes. Please add it.

Concerning the thermal analysis, I thing that it can be review. Some mistakes appear in the text, some temperatures are not in agreement with the figure. Is it also possible to describe more precisely the different step of weight loss?

I think that this paper can be publishing in Molecules after minor revisions.

Author Response

We also thank you for your useful comments and suggestions on our manuscript.

We have modified the manuscript accordingly, and the detailed corrections are listed below point by point. The changes in the text are marked in red.

In this paper the authors describe the synthesis, crystal structure and magnetic properties of two new Co-based compounds.

Concerning the crystal structure description, the figure must have the axes. Please add it.

Ans: We have added the axes to the figures. Concerning the thermal analysis, I thing that it can be review.

Some mistakes appear in the text, some temperatures are not in agreement with the figure. Is it also possible to describe more precisely the different step of weight loss?

Ans: We have further reviewed the thermal properties to describe the weight loss more precisely. Please see page 5. I think that this paper can be publishing in Molecules after minor revisions. Ans: We sincerely appreciate the reviewer’s comment.

Reviewer 2 Report

The paper by Chhetri et. al. described Co(II) supramolecular isomers based on flexible bis-pyridyl-bis-amide and angular dicarboxylate ligands and their magnetic properties. Complex 1 exhibits one-dimensional (1D) looped chain with L ligands penetrating into the middles of the squares, giving a new type of self-catenated net with the (42∙54)(4)2(5)2 topology, whereas complex 2 displays a 2-fold interpenetrated 2D net with the rare (42∙68∙8∙104)(4)2-2,6L1 topology. Overall, this is a nice piece of work and can be accepted for publication in Molecules with the following corrections.

The author should provide the colour code of crystal structure in Figures 1 and 2.

The author should make a brief description of what are the driving forces to generate supramolecular isomers?  

The author should assign characteristic peaks in the preparations section. And also describe IR spectral studies in the main text.

In the material and method section, Page 8 line 209-210, Emission spectra were obtained from a Hitachi F-4500 spectrometer (Hitachi, Tokyo, Japan). But we cannot find any emission spectral studies in the main text.

The Figure 3 caption “Thermal gravimetric analyses (TGA) curves for complexes 1” should be corrected as “Thermal gravimetric analyses (TGA) curves for complexes 1 and 2”.

In Figure 4, the simulated PXRD patterns of complex 1 have no meaning. It should be removed from Figure 4.

Some related work should be noted such as CrystEngComm, 2013, 15, 349 and Dalton Trans., 2015, 44, 6052

Author Response

Thank you for your useful comments and suggestions on our manuscript. We have modified the manuscript accordingly, and the detailed corrections are listed below point by point. The changes in the text are marked in red.

The paper by Chhetri et. al. described Co(II) supramolecular isomers based on flexible bis-pyridyl-bis-amide and angular dicarboxylate ligands and their magnetic properties. Complex 1 exhibits one-dimensional (1D) looped chain with L ligands penetrating into the middles of the squares, giving a new type of self-catenated net with the (42∙54)(4)2(5)2 topology, whereas complex 2 displays a 2-fold interpenetrated 2D net with the rare (42∙68∙8∙104)(4)2-2,6L1 topology. Overall, this is a nice piece of work and can be accepted for publication in Molecules with the following corrections.

Ans: We sincerely appreciate the reviewer’s comments.

The author should provide the colour code of crystal structure in Figures 1 and 2.

Ans: We have added the color codes to the figures of complexes 1 and 2.

Please see pages 3 and 4. The author should make a brief description of what are the driving forces to generate supramolecular isomers?

Ans: “The different structural types of 1 and 2 were most probably directed by the solvents, H2O and CH3OH, and the anions of the Co(II) salts, OAc- and BF4- that may exhibit the template effect on the structures of 1 and 2, respectively.”

Please see lines 71 – 73. The author should assign characteristic peaks in the preparations section. And also describe IR spectral studies in the main text.

Ans: We have assigned the IR characteristic peaks in the preparation section (page 8) and described in the main text. Please see lines 65 – 70, page 2.

In the material and method section, Page 8 line 209-210, Emission spectra were obtained from a Hitachi F-4500 spectrometer (Hitachi, Tokyo, Japan). But we cannot find any emission spectral studies in the main text.

Ans: We have removed this instrument, which was not used for this work, from the text.

The Figure 3 caption “Thermal gravimetric analyses (TGA) curves for complexes 1” should be corrected as “Thermal gravimetric analyses (TGA) curves for complexes 1 and 2”.

Ans: We have corrected the caption as suggested. Please see line 162, page 5, bottom.

In Figure 4, the simulated PXRD patterns of complex 1 have no meaning. It should be removed from Figure 4.

Ans: We have removed the simulation spectrum of 1 from Figure 4.

Please see page 6, top. Some related work should be noted such as CrystEngComm, 2013, 15, 349 and Dalton Trans., 2015, 44, 6052

Ans: We have cited these two works as references 9 and 10. Please see page 10.

Reviewer 3 Report

This is a straightforward and unambitious study of two compounds, which did not provide many challenges in synthesis even if the concept of the isomerization is neat.

It will interest coordination polymer chemists and can be accepted with minor revision.

The formula of 2 should be written with the lattice solvent outside of the square brackets. As currently presented the MeOH and H2O of two are coordinated and hence 1 and 2 are not isomers. In formulae, it should not be Co(II), but the II should be superscripted according to IUPAC.

In the experimental, the formulae for microanalysis data should be written doubled (i.e. as Co2 etc) as the headings have a Co2 formula

Author Response

Thank you for your useful comments and suggestions on our manuscript. We have modified the manuscript accordingly, and the detailed corrections are listed below point by point. The changes in the text are marked in red.

The formula of 2 should be written with the lattice solvent outside of the square brackets. As currently presented the MeOH and H2O of two are coordinated and hence 1 and 2 are not isomers. In formulae, it should not be Co(II), but the II should be superscripted according to IUPAC.

In the experimental, the formulae for microanalysis data should be written doubled (i.e. as Co2 etc) as the headings have a Co2 formula.

Ans: Corrected as suggested

Academic Editor's Comments:

I would like the authors to resolve a few minor issues.

  1. Add a caption for Scheme 1.
  2. Comment upon the separation between a pair of cobalt atoms in both compounds. Are they too long to signify direct metal-metal bonding interactions? Add also appropriate reference(s).
  3. Compound 2 contains solvent molecules of methanol and water. Please comment shortly about their location in the structure and their intermolecular interactions. Are they located in the cavities? Are crystals of 2 stable when taken out from the mother liquor?
  4. The manuscript will also require language editing. Please have your manuscript reviewed by a native speaker if possible.

Author Response

Thank you for your useful comments and suggestions on our manuscript. We have modified the manuscript accordingly, and the detailed corrections are listed below point by point. The changes in the text are marked in red.

1. Add a caption for Scheme 1.

Ans: A caption has been added to Scheme 1.

2. Comment upon the separation between a pair of cobalt atoms in both compounds. Are they too long to signify direct metal-metal bonding interactions? Add also appropriate reference(s).

Ans: As discussed in Cotton’s “Multiple bond between metal atoms”, the single-bonded dicobalt(II) complexes that are supported by the bridging ligands show Co-Co distances in the range from 2.2 to 2.4 Å, while those for the unsupported ones are about 2.8 Å. The Co-Co distances of complexes 1 and 2 in this manuscript that are supported by the carboxylate groups are 2.929(6) and 2.831(7) Å, respectively, are thus too long to be considered as bonds. The antiferromagnetic coupling can be the result of super-exchange interactions or direct dipole-dipole interactions.
“Cotton, F.A.; Murillo, C. A.; Walton, R. A. Multiple Bonds Between Metal Atoms, 3rd ed.; Springer Science and Business Media, Inc., New York, 2005” has been added to the text as reference [14].

3. Compound 2 contains solvent molecules of methanol and water. Please comment shortly about their location in the structure and their intermolecular interactions. Are they located in the cavities? Are crystals of 2 stable when taken out from the mother liquor?

Ans: “The water and methanol solvent molecules are located in the cavities of the interpenetrated framework. The water molecules interlink the independent nets through O-H---O hydrogen bonds to the amide oxygen atoms (O-H = 1.99 Å; OO--HH------O = 153.7O = 153.7o) of one net and to the carboxylate oxygen atoms of the other (O-H = 2.44 Å; OO--HH------O = 145.5O = 145.5o), while the methanol molecules link two water molecules, Figure S1.” Crystals of 2 were stable when taken out from the mother liquor. They remained stable when heated at 250 °C for two hours. (page 6, figure 4)

4. The manuscript will also require language editing. Please have your manuscript reviewed by a native speaker if possible.

Ans: We have done our best to edit this manuscript.